# Effectiveness of environment-aware AR interfaces on task performance in a workspace setting

Nattaon Techasarntikul
Osaka University
Japan

Koichi Owaki
Osaka University
Japan

Hideyuki Shimonishi
Osaka University
Japan

Figure 1: The goal of the research is to construct a digital twin of a human-robot coexisting environment. The virtual environment contains all the information of the real world obtained from sensors, including the positions of objects and the motions of humans and robots. In the current setting, humans are unaware of the approaching robot. However, the direction and distance of the robot from the human, which are calculated in the virtual environment, are transmitted to the physical environment. This information about the robot is then displayed as AR interfaces through the Hololens, allowing the humans to become aware of the robot.

## ABSTRACT

In a human-robot coexisting environment, understanding the user's intention would help the robot plan its work efficiency, and vice versa. Our work simulates a situation in which a user is working on a static position and a robot is operated in the same environment. While robots can quickly make decisions, the user who is currently focused on a task may be less aware of their surroundings and could run the risk of colliding with the robot. Conversely, the worker may also get sidetracked while monitoring the robot's motion, leading to inattentiveness and inefficiency in their task performance.

*Conference'17, July 2017, Washington, DC, USA*
© 2023 Association for Computing Machinery.
ACM ISBN 978-x-xxxx-xxxx-x/YY/MM...$15.00
https://doi.org/10.1145/nnnnnnn.nnnnnnn

Existing research has primarily focused on safe robot navigation, which could malfunction due to interruptions in control signals or other technical uncertainties; thus, it would be expected that humans are also notified of the robots' intentions in order to work with them efficiently. To address the problem of human-robot awareness, we conducted an experiment to examine whether a user could focus on their work while being aware of the robot. We developed Augment Reality (AR) glasses interfaces to inform the user of the nearby robot and conducted an experiment to study the impact of the level of information shown by AR interfaces on task performance and concentration. The results show that users require more head turning for checking the robot position when no information-aid is provided, nevertheless, it was also found that there was no significant difference in task performance when the user was multitasking non-skilled tasks. Among different levels of information on the AR interfaces, the users reported being more concentrated on their work when the system provided less information. We conclude that AR interfaces help increase concentration on the task based on the subjective-rated questionnaire results. The objective measurements reveal that the AR interfaces reduce the physical demand of turning and moving the body. However, the

different levels of information on each interface do not significantly affect the task performance in this study.

## CCS CONCEPTS

• **Human-centered computing → User studies**; **Mixed / augmented reality**.

## KEYWORDS

Augment Reality, AR glasses interface, human-robot awareness

**ACM Reference Format:**
Nattaon Techasarntikul, Koichi Owaki, and Hideyuki Shimonishi. 2023. Effectiveness of environment-aware AR interfaces on task performance in a workspace setting. In *Proceedings of ACM Conference (Conference'17)*. ACM, New York, NY, USA, 5 pages. https://doi.org/10.1145/nnnnnnn.nnnnnnn

## 1 INTRODUCTION

Nowadays, service robots are employed in many places as we can see them in an airport, department store, and restaurant. These robots work either automatically or by being tele-operated through a wireless network. Although the safety of human is the top concern in the operation of robots, accidents can still occur due to dynamic changes in the environment or delayed control signals. In this direction, controlling robots with a focus on human safety is a popular area of study. However, it is impossible to guarantee 100% safety in human-robot interaction (HRI) due to the potential errors in measurement or unpredictable circumstances [4], thus both robots and humans have to move very timidly. Therefore, it is expected that human should also self-aware of potential accidents in order to coexist and collaborate with robots much more efficiently.

Using the Augmented Reality (AR) interface on AR glasses allows a human to access assistive information hands-free while performing a task. Most of AR interfaces focus on either a path navigation [1] or a task assistant [5], leaving a research gap for interfaces that help the user become aware of surrounding risks while also allowing them to focus on their main task.

In this paper, we utilize the Digital-Twin (DT) concept of human-robot coexist (Fig. 1) to observe movement behaviours of humans and robots in the environment and inform the user of robot position through AR interface. DT is a digital replica of a physical environment with a data connection between them. It can be used for real-time simulation or future trend prediction to help control objects in the environment [2]. We hypothesize that in an environment where a worker needs to avoid a robot and perform a task at the same time, using an environment-aware AR interface would help them better concentrate on their task and lead to an increase in the task performance, compared to the non-aid interface.

## 2 RELATED WORK

Our study lies in the area of human navigation in AR and HRI, where the proposed system aims to help users become aware of the presence of nearby robots while focusing on their tasks.

In HRI, collision avoidance is more difficult for robots compared to humans, who can easily avoid each other [3]. Since many industrial and service robots are programmed to prioritize user safety, they may reduce their velocity or pause their current task until the user is out of the way. The low efficiency mode may last longer if the user is not aware of the robot's intention. However, it is impossible for users to know the robot's intention, such as where it will be moving, unless they are the operator. Displaying information about the robot's path or operation area could help the user avoid intersecting the robot's area, allowing the robot to continue working without interruptions [7]. As a result, both user safety and robot work efficiency could be improved. Walker et al. have developed several AR interfaces to show the drone moving intention [8]. Showing a virtual future path of a drone helps users to re-plan their task to avoid a collision with the drone faster than if they were observing the drone physically. A combination of visual and audio cues could help a user avoid entering dangerous zones [6]. However, the selection of the alert sound, duration, and volume progression can affect the user's aspect in a negative way.

In our study setting, when a user who is focused on their work and a robot approach from backside, displaying the robot's path or precaution area may not alert the user to the incoming robot. Therefore, we propose AR interfaces that indicate the robot's direction and distance from the user with the goal of allowing the user to avoid the robot while remaining focused on their work.

## 3 SYSTEM DESIGN AND AR INTERFACES

Our system simulates a situation in which a user is working on a static position and a robot is operated in the same environment as shown in Fig. 2. The user is wearing a HoloLens, a type of AR glasses, to receive information about the robot's position. Another HoloLens is placed on a robot to track its position in the environment. To share positions of multiple HoloLens in global coordinate (the current environment), Azure Spatial anchors function is enables and attached to both the user's and robot's HoloLens. The HoloLens transfer each other's position and orientation through a Photon Unity Networking (PUN) service. The direction of the robot relative to the user is obtained in real-time and displayed as a 3D directional arrow on the user's AR glasses. To notify the user of the approaching robot.

The AR navigation interfaces are designed to provide as little information as possible to avoid distracting the user from their primary task. Three types of interfaces show both the robot direction relative to the user and the distance in implicit and explicit ways. The *number* interface shows the exact numerical distance. The *inflate* interface resizes the 3D arrow based on the distance. The *sign* interface appears when the user is inside the critical area of the robot. The interface image captured from the HoloLens' view is shown in Fig. 2 (a).

## 4 EXPERIMENT SETUP

We provided an object separation task which can be found in manufacturing. Six types of plastic bricks, each with 10 pieces, for a total of 60 pieces were mixed before starting each task. The participant needs to separate these bricks into 6 containers in a predefined randomized order as shown in Fig. 2 (b). During the task, a robot is operated around the user, and the user needs to avoid collision with the robot while maintaining their task. We assumed that the robot's control signal may be interrupted, hence it may be unable to stop before colliding with the user, therefore the user needs to

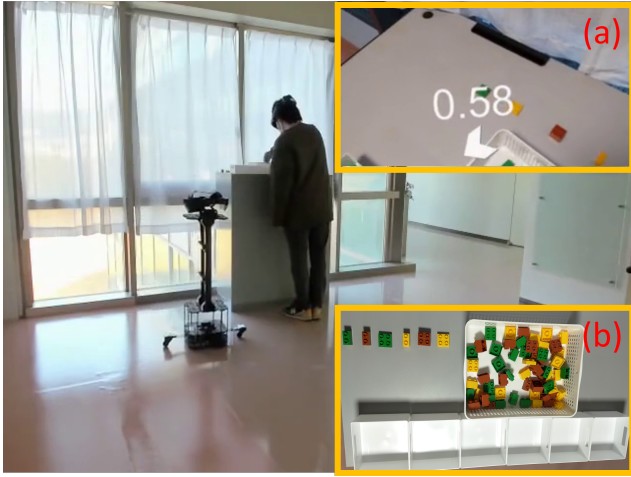

**Figure 2: Experiment scene. (a) a view through the HoloLens. (b) an object separating task that the user needs to complete.**

be self-aware of the robot. The navigation interfaces were designed to help the user be aware of the incoming robot.

We recruited 12 participants through our university web board. Average participant age was 20.67 (SD=1.45). To collect the head motion data, the participant needs to wear a HoloLens for all tasks, whether an interface is shown or not. They will be asked to perform the task five times. For the first and the last attempts, participants will arrange the bricks without any navigation aids as a baseline. For the middle three attempts, a randomized interface will be provided.

## 5 RESULTS

We measured the times that the participants took to complete each task and recorded their head positions and rotations directly from the HoloLens device. Additionally, after completing each task, we asked the participants to rate their level of concentration during the task. A repeated measure Analysis of Variance (ANOVA) method is used to evaluate whether there are significant differences between groups in parametric measurements. Then a post-hoc analysis using pairwise t-tests with Holm's adjustment is conducted to determine which pairs of groups exhibit a significant difference.

A box and whisker plot is used to visualize the spread of measured data. The bar in the middle of the box represents the median value, while the lower and upper whiskers indicate the minimum and maximum values, respectively. Any data points below or above the whiskers are considered outliers. We indicate a significance level of $p < 0.001$, $p < 0.01$, and $p < 0.05$ using asterisks ***, **, and *, respectively, between each set of bars in our results plots.

### 5.1 Task completion time

An average time used to complete the task for each navigation interface is shown in Fig. 3 (a). The ANOVA test results indicate that there was no significant difference between the baselines and proposed interfaces ($F = 0.470$, $p = 0.757$). Furthermore, we also examined the potential influence of the learning effect on task order, and the ANOVA analysis indicated that there was no noteworthy

difference in the time taken to complete the task based on the sequence in which the tasks were performed ($F = 0.464$, $p = 0.700$).

**Table 1: Average task completion time (seconds)**

|      | none first | number | inflate | sign | none last |
|------|-----------|--------|---------|------|-----------|
| Mean | 296.50 | 280.75 | 270.58 | 265.91 | 255.25 |
| SD | 73.03 | 90.79 | 82.84 | 73.47 | 73.58 |

### 5.2 Task Concentration

Fig. 3 (b) shows the user's self-rated level of concentration (5-Point Likert Scale) on the task while using each interface. A non-parametric Friedman test was used to analyze the questionnaire results and a significant in task concentration ($\chi^2 = 21.746$, $p < 0.001$) is found. While a post-hoc analysis using pairwise Wilcoxon tests with Holm's adjustment did not find any differences between pairs, we instead used Hommel's adjustment only on this rating result to find identify the differences. For comparison, we report both the Hommel and Holm results as follows:

- *none first* and *number* ($p = 0.040$ for Hommel, $p = 0.061$ for Holm)
- *none first* and *inflate* ($p = 0.032$ for Hommel, $p = 0.053$ for Holm)
- *none last* and *inflate* ($p = 0.040$ for Hommel, $p = 0.061$ for Holm)
- *none last* and *sign* ($p = 0.041$ for Hommel, $p = 0.061$ for Holm)

**Table 2: User rating on task concentration**

|      | none first | number | inflate | sign | none last |
|------|-----------|--------|---------|------|-----------|
| Mean | 1.75 | 3.58 | 3.58 | 3.66 | 2.08 |
| SD | 0.96 | 1.16 | 0.90 | 1.20 | 0.66 |

### 5.3 Movement

The average and total movement during performing each task are shown in Fig. 3 (c) and (d), respectively. ANOVA found a significant difference on average movement ($F = 3.782$, $p < 0.01$) but did not find a significant difference on total movement ($F = 1.344$, $p = 0.265$). On the average movement, a post-hoc analysis revealed differences between *none first* and *inflate* ($p = 0.014$).

**Table 3: Average movement (centimeters/second)**

|      | none first | number | inflate | sign | none last |
|------|-----------|--------|---------|------|-----------|
| Mean | 13.90 | 11.26 | 10.74 | 11.29 | 12.69 |
| SD | 2.44 | 2.43 | 2.18 | 2.45 | 2.22 |

**Table 4: Total movement (centimeters)**

|      | none first | number | inflate | sign | none last |
|------|-----------|--------|---------|------|-----------|
| Mean | 4150.64 | 3289.25 | 2945.17 | 3087.33 | 3287.82 |
| SD | 1513.67 | 1578.34 | 1343.85 | 1408.89 | 1271.26 |

## 5.4 Rotation

The average and total rotation during performing a task using each navigation interface is shown in Fig. 3 (e) and (f), respectively. ANOVA found a significant difference on average rotation ($F = 8.251$, $p < 0.0001$) and total rotation ($F = 4.721$, $p < 0.01$).

On the average rotation, a post-hoc analysis revealed significant differences between the *none first* baseline and other proposed interfaces, including *number* ($p = 0.005$), *inflate* ($p = 0.005$), and *sign* ($p = 0.020$). Significant differences between the *none last* baseline and other proposed interfaces are *number* ($p = 0.001$), *inflate* ($p = 0.001$), and *sign* ($p = 0.005$).

On the total rotation, a post-hoc analysis found significant differences between the first task's non-aid baseline and the proposed interfaces as follows: *none first* and *number* ($p = 0.034$), *none first* and *inflate* ($p = 0.012$), and *none first* and *sign* ($p = 0.037$).

**Table 5: Average rotation (degrees/second)**

|      | none first | number | inflate | sign  | none last |
|------|-----------:|-------:|--------:|------:|----------:|
| Mean | 31.63      | 22.13  | 22.25   | 23.64 | 33.21     |
| SD   | 7.59       | 5.42   | 6.14    | 6.65  | 7.31      |

**Table 6: Total rotation (degrees)**

|      | none first | number  | inflate | sign    | none last |
|------|-----------:|--------:|--------:|--------:|----------:|
| Mean | 9219.73    | 6132.22 | 5726.05 | 6212.16 | 8406.30   |
| SD   | 3243.01    | 2155.17 | 1806.61 | 2228.75 | 3096.32   |

## 5.5 Correlation

A Pearson correlation analysis was used to identify any correlations among the measures of task performing time, head rotation, and task concentration rating. Positively correlations were found between task performing time and head rotation with correlation coefficient $r = 0.59$ ($p < 0.001$). A moderately negative correlation was found between the user-rated concentration time and head rotation ($r = -0.363$, $p = 0.004$).

## 6 CONCLUSION AND DISCUSSION

In this paper, we presented and evaluated AR interfaces which aim to help the user become aware of a nearby robot while performing a task. Although there was no significant difference found in task completion time between using and not using AR interfaces, all of the AR interfaces significantly reduced the total time of head rotation compared to the non-assist conditions. This is because these AR interfaces informed the users how much the robot is closer to them, allowing them to avoid unnecessary head rotation to check the robot's position. Most participant prefers using the *sign* interface as it appears only when the robot get close to the user, thus distracting the users less from performing the task.

By investigating HoloLens measurement data of each participant, we found that the users took a bigger steps to avoid the approaching robot and often turned back to observe the robot's position when there was no robot information present. In cases that the robot was not approaching the user, using *number* and *sign* allowed the

participant to continue performing the task without rotating themselves to check the robot. However, participants still often turned back when using the *inflate*, which did not provide the user with accurate information about the distance to the robot. In the case that the robot is approaching near to the user, *number* helps the participants aware that the robot is coming and made them avoid the robot immediately without turning to check the robot. However, with the *inflate* and *sign* interfaces, users needed to turn back to check how far the robot was from them. Correlation analysis revealed that task performing time increases as head rotation increases. Moreover, with the increase of head rotation the user-rated concentration decreases. This can imply that constantly checking and avoiding a robot leads to a decrease of task concentrate and task performance.

We found that the task performance did not improve. One possible explanation could be the simplicity of the task itself (separating distinguishable object), which it would need to be further studied. A further investigation using more complex tasks, such as separating nuts by size or picking similar objects, would provide a more comprehensive result. Accordingly, our AR environment-aware interfaces can help the users focus on their work by taking care of the robot position monitoring and informing the user of the risk of collision, without requiring them to regularly check the robot.

## ACKNOWLEDGMENT

This work was supported by MIC under a grant entitled "R&D of ICT Priority Technology (JPMI00316)".

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

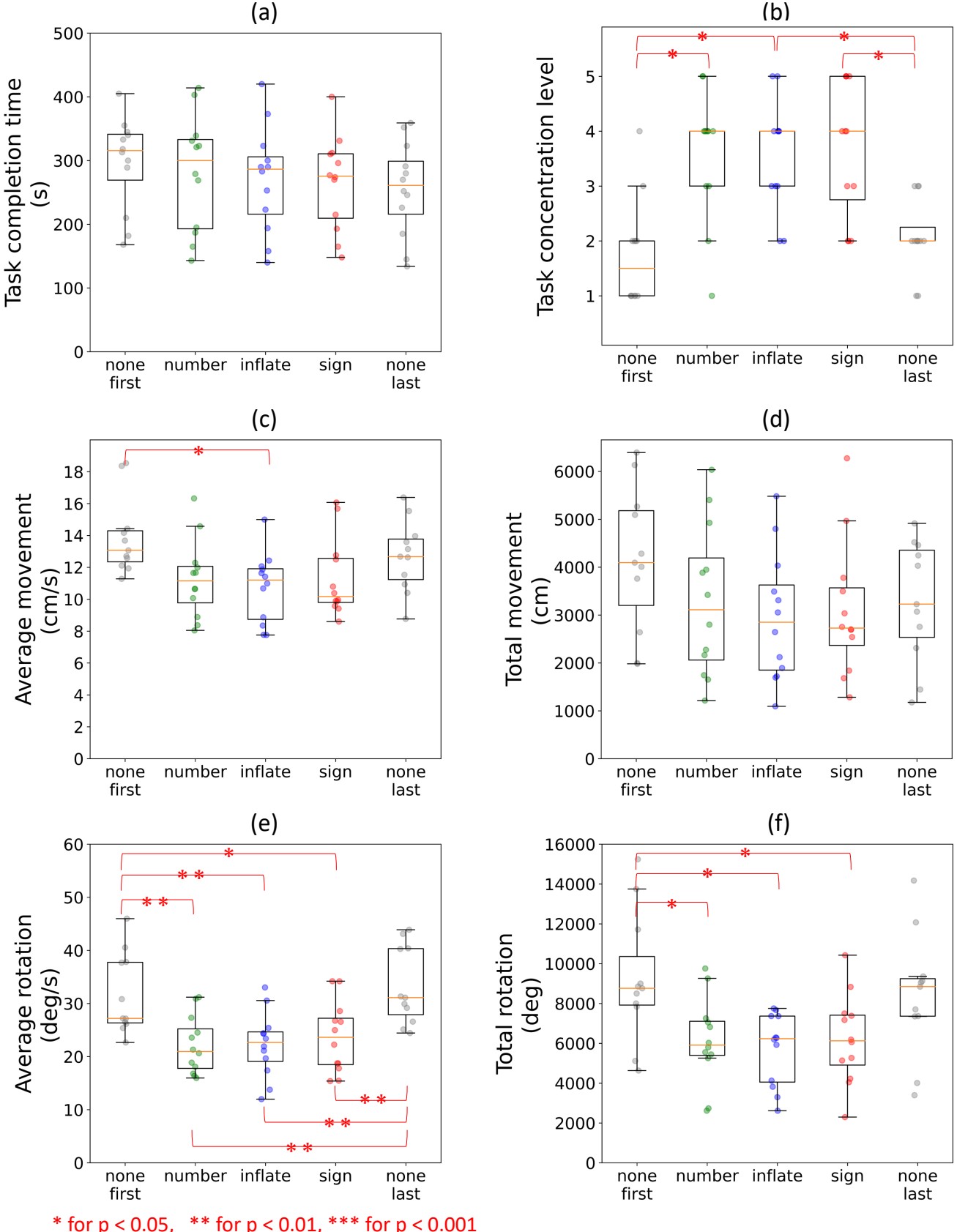

Figure 3: Experiment results.