# OpenReview forum: "Effectiveness of environment-aware AR interfaces on task performance in a workspace setting"
_humanrobotinteraction.org/HRI/2023/Workshop/VAM-HRI — VAM-HRI 2023 Oral_

### Official Review · Program_Chairs · 2023-02-25
**Accept**

**Rating:** 6
**Confidence:** 5

**Review:**

Review 1:

This paper describes the use of AR signaling for human-robot situational awareness as compared to no signaling. They conducted a study (n=12) where participants wear the Hololens to complete a set of 5 sequential brick-based organizational tasks (no signals, 3 types of signals, no signals). They report users stating a higher ability to concentrate on their task when no AR signaling was used but an increase in needed head turning. There was no significant change in task completion time between conditions.


I recommend a marginal acceptance.


Comments:
* “Effective of environment-aware AR interfaces on task performance in a workspace setting” -> I believe you mean “Effectiveness of…”
* “While robots can quickly make decisions, the user who is currently focused on a task may be less unaware” -> less aware*
* Could you put a figure/diagram of the tasks the users were doing? Why did you have participants wear the hololens for all tasks?
* Holm’s adjustment is great to see
* “Due to the simple nature of separating distinguishable objects, the task performance does not decrease.” -> you cannot claim this unless it was controlled for. You can speculate on it i.e., “Task performance did not improve. One possible explanation could be the simplicity of the task itself but this would need to be further studied.” - you do this in the following sentence but would need to state that the simple task leading to the result is a speculation in the first sentence
* Did not randomize/counter balance the order of no signal exposure (https://www.scribbr.com/methodology/within-subjects-design/) leading to possible ordering effects: https://www.statology.org/order-effects/ ; You did randomize the signals themselves but using no signal baseline at the beginning and end shows issues (For example, if you find a significant result between beginning and signal A but then do not find the same for signal A vs the end, then that could show a possible order effect). Please address this or I am happy to be corrected as a reviewer
* Label your y axis as opposed to putting the labels in the titles
* Figure 1 caption is not adding very much as I’m a bit confused on the arrows in the middle. Could you please elaborate either in the diagram or in the figure caption what it is exactly supposed to show?

Review 2:

This paper presents a within-subjects user study comparing three different types of AR-based proximity alerts warning a user of a potential collision with a robot operating behind the human versus a baseline of no alert. The three alerts all show a 3D arrow pointing to the robot. The “number” condition shows the distance to the robot. The “inflate” condition resizes the 3D arrow based on proximity. And the “sign” condition displays a specific alert only when the robot enters a safety envelope around the human. These interfaces were compared on a variety of metrics as the human conducted a task with a robot behind them: task completion time, rated ability to concentrate on the task, and head movement and rotation. While no effect was found for task completion time, the AR interfaces were generally associated with higher rated task concentration and less head rotation compared to the control. Effects between the AR conditions were not found.

Strengths:
- The statistical analysis presented is quite thorough and detailed, and seems to be conducted properly with respect to the within-subject design, which is always good to see in HRI research. All in all, the presentation of the results of the experiment is well done.
- The experiment gets at an interesting problem of alerting humans of robot proximity while they are looking away from the robot, and compares three highly plausible AR designs for such an alert interface. This work is definitely of interest to the VAM-HRI research community.

Weaknesses:
- Figure 1 is confusing: since this work is about AR interfaces for alerting a human collaborator about proximity to a robot, the left image seems more relevant. The rest of the image, with the various arrows and virtual human-robot environment, are not described in the paper beyond a passing reference to the digital twin concept. Relatedly, it is unclear exactly what the digital twin concept has to do with the content of this paper, so should either be expanded on or removed. Either way, Figure 1 should have a more detailed caption than “goal of the research” to reduce confusion.
- While Figure 2 is possible to interpret, the significance lines do get very cluttered in image (e). I would also suggest repeating the visual convention (* for p < 0.05, etc.) in the caption for Figure 2.
- I don’t understand the choice of having “no visualization” always presented 1st and 5th, rather than simply including it among the randomly ordered conditions. What advantage is there for this experimental design that supersedes the imbalance in data collection and clear ordering effects? It seems strange to present statistical significance measures of these repeated conditions separately.
- The conclusion and discussion section is rather hard to follow, since numerous findings are presented in paragraph form one after the other. Maybe a few bulleted or numbered “key findings” could be included here to make the contribution stand out to readers. That said, many of the findings comparing specific interfaces are merely speculative, given the lack of significant effect between AR conditions.
- There are numerous grammatical errors throughout the paper, including in the title: I’m thinking it should read “Effect of” rather than “Effective of”. Would suggest editing for clarity of language before submitting the camera-ready version.

Despite lingering issues regarding experimental design and presentation, this is a potentially valuable work, of direct interest to the VAM-HRI research community. I believe that, after taking into consideration some of the reviewer feedback for the camera-ready version, it would benefit from inclusion in the program for VAM-HRI. I therefore recommend acceptance.

---

### Decision · Program_Chairs · 2023-03-02

Accept (Oral)